**RESEARCH**

# Differential richness inference for 16S rRNA marker gene surveys

M. Senthil Kumar[1,2*], Eric V. Slud[3,4], Christine Hehnly[5,6†], Lijun Zhang[5†], James Broach[5,6], Rafael A. Irizarry[1,2], Steven J. Schiff[7†] and Joseph N. Paulson[6,8*†]

†Steven J. Schiff and Joseph N. Paulson are co-senior authors.

†Christine Hehnly and Lijun Zhang contributed equally to this work.

*Correspondence:
senthil@ds.dfci.harvard.edu;
jpaulson@psu.edu

[1] Department of Data Science, The Dana-Farber Cancer Institute, Boston, MA, USA
[2] Department of Biostatistics, Harvard T.H. Chan School of Public Health, Boston, MA, USA
[3] Department of Mathematics, University of Maryland, College Park, MD, USA
[4] Center for Statistical Research and Methodology U.S. Census Bureau, Suitland, MD, USA
[5] Penn State Institute for Personalized Medicine, The Pennsylvania State University College of Medicine, Hershey, PA, USA
[6] Department of Biochemistry and Molecular Biology, The Pennsylvania State University College of Medicine, Hershey, PA, USA
[7] Department of Neurosurgery, Yale University, New Haven, CT 06510, USA
[8] Department of Data Sciences, Product Development, Genentech, South San Francisco, CA, USA

## Abstract

**Background:** Individual and environmental health outcomes are frequently linked to changes in the diversity of associated microbial communities. Thus, deriving health indicators based on microbiome diversity measures is essential. While microbiome data generated using high-throughput 16S rRNA marker gene surveys are appealing for this purpose, 16S surveys also generate a plethora of spurious microbial taxa.

**Results:** When this artificial inflation in the observed number of taxa is ignored, we find that changes in the abundance of detected taxa confound current methods for inferring differences in richness. Experimental evidence, theory-guided exploratory data analyses, and existing literature support the conclusion that most sub-genus discoveries are spurious artifacts of clustering 16S sequencing reads. We proceed to model a 16S survey's systematic patterns of sub-genus taxa generation as a function of genus abundance to derive a robust control for false taxa accumulation. These controls unlock classical regression approaches for highly flexible differential richness inference at various levels of the surveyed microbial assemblage: from sample groups to specific taxa collections. The proposed methodology for differential richness inference is available through an R package, *Prokounter*.

**Conclusions:** False species discoveries bias richness estimation and confound differential richness inference. In the case of 16S microbiome surveys, supporting evidence indicate that most sub-genus taxa are spurious. Based on this finding, a flexible method is proposed and is shown to overcome the confounding problem noted with current approaches for differential richness inference.

Package availability: https://github.com/mskb01/prokounter

**Keywords:** Microbiome, Spurious, Surveys, False discoveries, Species misclassification, Richness, Differential richness

## Background

Clinically relevant health outcomes are often accompanied by changes in the diversity of associated microbial communities. For instance, decreased gut microbiome diversity accompanies childhood diarrhea [1] and enteric infections [2] and has been

shown to predict the onset of infant type I diabetes [3]. Distinct intra-tumoral microbial diversity levels are associated with cancer sub-types [4–6]. Thus, inferring disease associated changes in microbiome diversity metrics is useful for characterizing disease pathology and progression.

Among the various diversity measures, *richness* quantifies the number of taxonomic groups in a community [7, 8]. Changes in species richness of biological communities have informed key environmental management practices that are relevant to public health and well-being [7–23]. Of the technologies available for characterizing microbial communities, 16S rRNA (ribosomal RNA) gene surveys are widely adopted for their high throughput and low cost. As a broad screening tool, they largely avoid the need for laborious culturing of microbes. This makes them especially attractive for deriving health metrics based on the microbiome.

In this work, we focus on inferring changes in richness of microbial communities between sample groups (i.e., differential richness) with 16S survey data.

To infer differential richness, one first estimates richness for the specific communities of interest in each survey sample. The estimated values are then compared between sample groups with either fixed or mixed effects models, or with non-parametric statistical tests, possibly adjusting for sampling effort [24–27]. There are two types of sample-level estimates of richness. *Observed* richness refers to the number of taxa observed in a sample. *Asymptotic* richness is obtained by adding an estimate of the number of unobserved taxa to the number of observed taxa. Approaches to estimate asymptotic richness vary, but often assume that relatively uncommon taxa are the most informative [28]. Both types of richness estimates enable valid comparisons among *macro*-ecological communities [24, 25, 28–30].

However, direct application of the aforementioned richness estimates and comparisons to 16S microbiome data would ignore the plethora of uncommon and spurious taxa that inflate observed richness estimates in 16S survey data [27, 31–34]. When this artificial inflation in observed richness is ignored, we find that differential abundance of detected taxa confounds current methods for differential richness inference. The problem is severe when between-sample richness comparisons are made at lower taxonomic levels, e.g., genus. Thus, direct application of classical methods to microbiome differential richness inference is unreliable.

Attempts to overcome sequencing noise have been made. Chiu and Chao [35], noting that singleton taxa are highly susceptible to sequencing noise, establish an improved estimator for counting singleton taxa by relying on more abundant taxa (also see Willis [36]). However, the estimator is often numerically undefined at lower levels of the taxonomy, and still takes most of the observed richness at face value.

Our results indicate that the observed frequencies of spurious taxa are determined by the output abundances of input sequences and thus need not be restricted to singleton frequencies alone. We therefore aimed to develop a flexible differential richness inference procedure for 16S microbiome surveys—one that would not only allow investigators to seek sample-wide richness changes across experimental groups (as is commonly done in modern metagenomics) but also within genera or taxa collections of any particular interest, while accounting for false taxa accumulations.

The paper is divided into several sections. The "Most sub-genus taxa in 16S surveys are likely technical artifacts" section, based on our own experiments and exploratory data analyses guided by theory, presents multiple lines of evidence supporting the view that most sub-genus taxa currently identified in 16S surveys are spurious. This allows us to exploit within-genus taxa accumulation data to derive a robust control for false taxa accumulations (the "Methods" section). The "Spurious taxa confound differential richness inference" section illustrates the confounded differential richness inferences arising from current methods, when detected taxa exhibit a net non-zero relative abundance fold change between sample-groups. The "Prokounter enables flexible differential richness inference" section applies the proposed procedure (*Prokounter*) to a variety of datasets and illustrates the value that differential richness inferences at lower taxonomic levels add to clinical and public health-related microbiome data analyses. For example, application of *Prokounter* to a gut microbiome survey of a traveling individual [2] identifies the invading genera with increased richness in member taxa, during and after an enteric infection.

## Results

### Most sub-genus taxa in 16S surveys are likely technical artifacts

16S surveys reconstruct target microbial populations by clustering sequencing reads. Spurious microbial taxa occur when the clustering procedure's error model fails to capture the entirety of sequence variation induced by the technical steps in 16S sequencing. These steps include, but are not limited to, PCR amplification of 16S material and sequencing (Fig. 1A).

To identify the major parameters underlying false taxa accumulations, we mathematically model the nucleotide substitution errors introduced by a chain of PCR

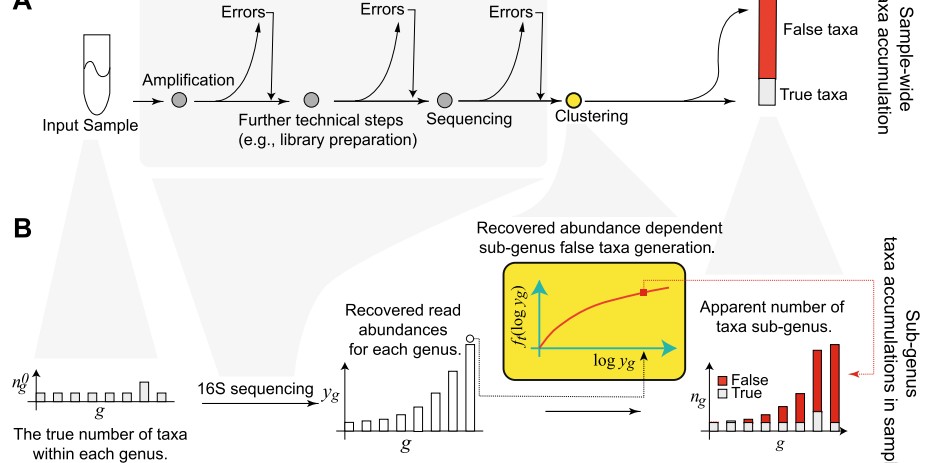

**Fig. 1** Within-genus false taxa accumulation structure. **A** Sequences in input samples are subjected to various technical steps during 16S sequencing (gray shade). The output reads from 16S sequencing are clustered for sequence similarity using a methodology of choice. Of the number of taxa (clusters) thus reconstructed, some are true, i.e., equal in sequence to those in the input sample, the rest are spurious i.e., false (red). **B** For every genus, the accumulation is determined as a function of its recovered abundances. Notation: $n_g^0$ the respective true number of taxa associated (true richness), $y_g$ the genus recovered abundance, $f_t(\cdot)$ the abundance dependent technical component driving false taxa accumulations within-genus

amplification and sequencing processes allowing for back mutations (Additional File 1 [37–41]). Under reasonable assumptions, we find that the rate of falsely classifying an error variant of a source sequence (type I error) using a priori fixed sequence similarity thresholds strongly increases with the source sequence's recovered (i.e., output) abundance. The average recovered abundance is multiplicative in the source sequence's apparent input abundance and the total sampling depth (Additional File 1). Thus, false sequence clustering decisions, and hence the resulting false clusters, increasingly accumulate with the true source sequence's recovered abundance and not necessarily sample depth. We therefore identify a mechanism through which spurious clusters of sequences are increasingly identified as microbial taxa, regardless of the underlying biological reality.

Given the empirical observation that 16S genetic segments are mostly limited in resolution to prokaryotic genera [42–49], we explored within-genus taxa accumulations (i.e., the number of detected sub-genus taxa as a function of recovered genus abundances), in several publicly available 16S surveys. In general, we expect genera to vary in their true richness and the relative abundances of member taxa. This must accordingly induce biological variation in the genus-specific taxa accumulation patterns. However, this expectation did not broadly hold in the several microbiome surveys analyzed here. Within-genus taxa accumulation patterns were highly concordant for several genera within study (Fig. 2A, Additional File 2: Fig. S1-S3). Relative to the number of detected genera, which ranged from 60 to 400 across studies, a clustering analysis indicates that within-genus taxa accumulation data supports only 2–8 distinct accumulation patterns in each study (Additional File 3: Table S1). Multiple dominant genera can be clustered to the same accumulation pattern. In addition, relative to study specific covariates, a robust trend estimate of the within-genus taxa accumulation data explains the bulk of the variation in genus-specific and sample-wide taxa accumulations (Tables 1 and 2) in each study. Similar qualitative conclusions follow when genus recovered abundance is used as a predictor, instead of an estimated trend (Additional File 3: Tables S2-S3). Finally, these qualitative and quantitative attributes of the accumulation patterns were obtained regardless of the 16S clustering approach used (Tables 1 and 2). These results indicate a strong within-study regularity in observed taxa accumulations across genera and sample groups—as if most genera have similar taxa richness and evenness—suggesting a likely technical origin.

### Single colony experiment

To further verify these conclusions, we conducted a 16S sequencing experiment on a target *Pseudomonas aeruginosa* population. The experimental sample was by itself overnight derived from a single *P. aeruginosa* colony (Additional File 1). In a series of experimental samples, we varied both the input abundance of Pseudomonas cells and the PCR amplification cycles. Our mathematical model (Additional File 1), which tracked the probability distribution of cell division induced nucleotide substitutions over generations, indicates that under no selection pressure, we can expect one biological 16S genotype in our input. An upper bound on the number of our input taxa is given by the number of 16S genes generally found within the *Pseudomonas* genus ($\sim 4$), times two for taxa clusters corresponding to forward and reverse complement

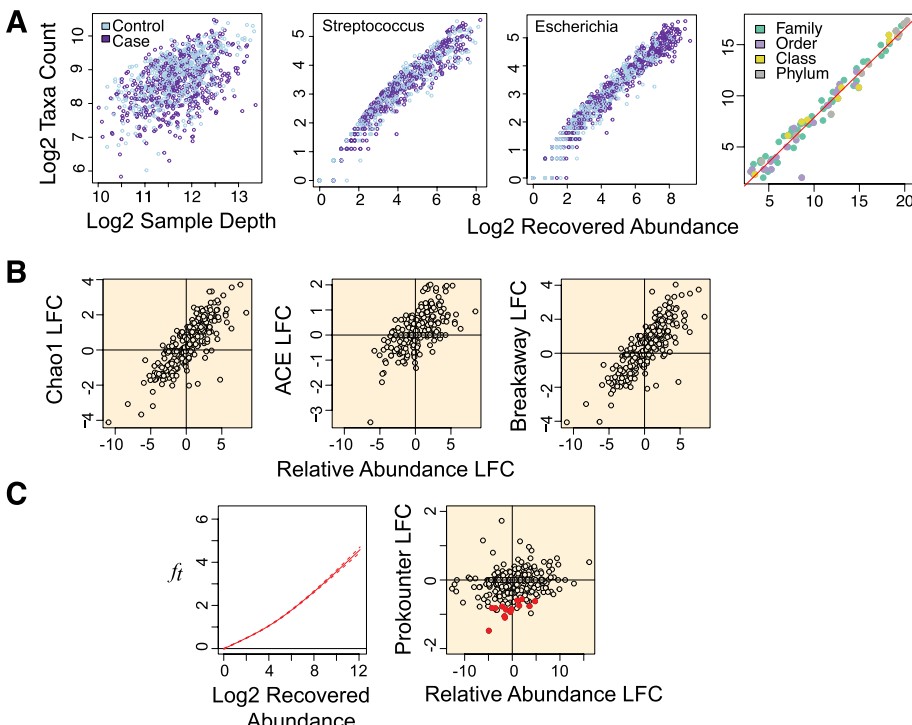

**Fig. 2** Concordant taxa accumulations across genera, confounded differential richness inference, and the Prokounter strategy. **A** Sample-wide taxa accumulations are visualized with respect to sample depth (left). Within-genus taxa accumulations are visualized with respect to the total recovered genus abundances for two gener a, i.e., the sum of the abundances of all taxa within the genus. **B** Differential richness log-fold changes (LFC, *y*-axis) track differential relative abundance fold changes (LFC, *x*-axis) in the waste-water treatment survey. **C** Prokounter exploits within-genus accumulation data to model false taxa accumulation rates. When exploited in a standard Poisson regression setting, the resulting differential richness fold changes are uncorrelated with genus-wide differential abundance statistics (right). Dashed lines represent confidence intervals. Points colored in red are the genus-specific differential richness inferences for the waste-water treatment survey

strands. What we found was a rather different representation, rich with low abundant and poorly replicating taxa: the total numbers of observed *Pseudomonas* taxa were 1050 and 300 for clustering methods based on sequence similarity with respective thresholds of 99% and 97% and 90 for a probabilistic clustering method (Dada2). The bulk of the newly identified *Pseudomonas* taxa preferentially contributed to the low frequency regime of the taxa abundance histogram (Additional File 2: Fig. S4), suggesting that they are likely clusters of rare, erroneous 16S sequencing reads generated during amplification and sequencing. Notably, taxa within-Pseudomonas, despite having a noisy occurrence with respect to amplification cycles and input cells (Fig. 3A), accumulated along the Pseudomonas genus recovered abundance axis in a clear, robust fashion (Fig. 3B). As expected, the stricter the sequence similarity threshold, the stronger the rate of taxa accumulations along the recovered abundance axis (Fig. 3B). Furthermore, taxa accumulations from several detected genera followed quantitatively similar patterns (Fig. 3C, Tables 1 and 2). From prior experiments in our laboratory and from control samples, we know *Pseudomonas* lab contaminants have very weak relative abundances. Restricting the above analysis to only

**Table 1** Relative to study variables, within-genus taxa accumulation trends capture bulk of the systematic variation in 16S surveys' *genus-specific* taxa accumulations. For each 16S survey mentioned in column 1, the year of publication is listed in column 2, the partial 16S segment targeted, machine technology and sequence clustering approach used are specified in column 3. FST.x refers to sequence clustering at an a priori fixed sequence similarity threshold of x%. McFadden's pseudo-$R^2$ for explaining genus-specific taxa accumulations with two negative binomial regressions (NB) are listed in columns 4 and 5. The fourth column is obtained when the NB regression includes within-genus taxa accumulation trend ($\hat{f}_R$(), Methods) alone as predictor. The fifth column additionally includes the genus identifier, total sample depth, and experimental design matrix for each dataset as predictors (methods). Corresponding Akaike Information Criteria (AIC) are listed in columns 6 and 7

| Dataset | Year | 16S segment, Sequencing & Clustering | Pseudo $R^2_{trend}$ | Pseudo $R^2_{trend+design}$ | $AIC_{trend}$ | $AIC_{trend+design}$ |
|---|---|---|---|---|---|---|
| Mouse[87] | 2009 | V2, 454, FST.97 | 96.82% | 98.68% | $1.250 \times 10^4$ | $1.0273 \times 10^4$ |
| Diarrhea[88] | 2014 | V12, 454, FST.99 | 98.52% | 98.96% | $1.2323 \times 10^5$ | $1.1455 \times 10^5$ |
| Time series[2] | 2014 | V4, GAIIx, FST.97 | 94.43% | 98.46% | $3.9401 \times 10^4$ | $3.0813 \times 10^4$ |
| Wastewater[55] | 2018 | V34, MiSeq, FST.99 | 91.70% | 95.51% | $3.7400 \times 10^4$ | $2.3682 \times 10^4$ |
| MBQC-HLB$^{(97)}$[27] | 2017 | V4, MiSeq, FST.97 | 97.15% | 98.66% | $1.0710 \times 10^5$ | $9.2275 \times 10^4$ |
| MBQC-HLB$^{(99)}$[27] | 2017 | V4, MiSeq, FST.99 | 98.67% | 99.29% | $1.089 \times 10^5$ | $9.6674 \times 10^4$ |
| MBQC-HLB$^{(D)}$[27] | 2017 | V4, MiSeq, Dada2 | 61.03% | 81.72% | $3.82 \times 10^4$ | $3.3877 \times 10^4$ |
| PIH100$^{(97)}$[56] | 2020 | V12, MiSeq, FST.97 | 94.04% | 97.32% | $1.5740 \times 10^4$ | $1.3858 \times 10^4$ |
| PIH100$^{(99)}$[56] | 2020 | V12, MiSeq, FST.99 | 97.66% | 98.90% | $1.7351 \times 10^4$ | $1.5739 \times 10^4$ |
| PIH100$^{(D)}$[56] | 2020 | V12, MiSeq, Dada2 | 88.37% | 91.01% | $8.8630 \times 10^3$ | $9.0263 \times 10^3$ |
| Pseudomonas$^{(97)}$[89] | 2021 | V12, MiSeq, FST.97 | 97.49% | 98.92% | $3.0762 \times 10^3$ | $2.8235 \times 10^3$ |
| Pseudomonas$^{(99)}$[89] | 2021 | V12, MiSeq, FST.99 | 98.71% | 99.46% | $3.3810 \times 10^3$ | $3.0717 \times 10^3$ |
| Pseudomonas$^{(D)}$[89] | 2021 | V12, MiSeq, Dada2 | 89.26% | 93.77% | 1881.62 | 1889.10 |

**Table 2** Relative to study variables, within-genus taxa accumulation trends capture bulk of the systematic variation in 16S surveys' *sample-wide* taxa accumulations. For each 16S survey mentioned in column 1, the year of publication is listed in column 2, the partial 16S segment targeted, machine technology and sequence clustering approach used are specified in column 3. FST.x refers to sequence clustering at an a priori fixed sequence similarity threshold of x%. McFadden's pseudo-$R^2$ for explaining sample-wide taxa accumulations with two negative binomial regressions (NB) are listed in columns 4 and 5. The fourth column is obtained when the NB regression includes within-genus taxa accumulation trend ($\hat{f}_R 0$, Methods) alone as predictor. The fifth column additionally includes the total sample depth, and experimental design matrix for each dataset as predictors (methods). Corresponding Akaike Information Criteria (AIC) are listed in columns 6 and 7

| Dataset | Year | 16S segment, Sequencing& Clustering | Pseudo $R^2_{trend}$ | Pseudo $R^2_{trend+design}$ | $AIC_{trend}$ | $AIC_{trend+design}$ |
|---|---|---|---|---|---|---|
| Mouse[87] | 2009 | V2, 454, FST.97 | 99.91% | 99.92% | $1.2971 \times 10^3$ | $1.2853 \times 10^3$ |
| Diarrhea[88] | 2014 | V12, 454, FST.99 | 99.94% | 99.94% | $1.2508 \times 10^4$ | $1.2406 \times 10^4$ |
| Time series[2] | 2014 | V4, GAIIx, FST.97 | 99.95% | 99.95% | $2.0972 \times 10^3$ | $1.9238 \times 10^3$ |
| Wastewater[55] | 2018 | V34, MiSeq, FST.99 | 99.96% | 99.97% | $6.7384 \times 10^2$ | $6.3222 \times 10^2$ |
| MBQC-HLB$^{(97)}$[27] | 2017 | V4, MiSeq, FST.97 | 99.98% | 99.99% | $2.5052 \times 10^3$ | $2.4587 \times 10^3$ |
| MBQC-HLB$^{(99)}$[27] | 2017 | V4, MiSeq, FST.99 | 99.992% | 99.993% | $2.7456 \times 10^3$ | $2.6955 \times 10^3$ |
| MBQC-HLB$^{(D)}$[27] | 2017 | V4, MiSeq, Dada2 | 99.86% | 99.03% | $1.6748 \times 10^3$ | $1.5003 \times 10^3$ |
| PIH100$^{(97)}$[56] | 2020 | V12, MiSeq, FST.97 | 98.96% | 99.07% | $1.2430 \times 10^3$ | $1.1989 \times 10^3$ |
| PIH100$^{(99)}$[56] | 2020 | V12, MiSeq, FST.99 | 99.94% | 99.95% | $1.4471 \times 10^3$ | $1.3988 \times 10^3$ |
| PIH100$^{(D)}$[56] | 2020 | V12, MiSeq, Dada2 | 99.73% | 99.73% | $9.8011 \times 10^2$ | $9.8415 \times 10^2$ |
| Pseudomonas$^{(97)}$[89] | 2021 | V12, MiSeq, FST.97 | 99.94% | 99.95% | $3.0641 \times 10^2$ | $2.8237 \times 10^2$ |
| Pseudomonas$^{(99)}$[89] | 2021 | V12, MiSeq, FST.99 | 99.97% | 99.98% | $3.1263 \times 10^2$ | $2.9741 \times 10^2$ |
| Pseudomonas$^{(D)}$[89] | 2021 | V12, MiSeq, Dada2 | 99.83% | 99.84% | $1.8723 \times 10^2$ | $1.9202 \times 10^2$ |

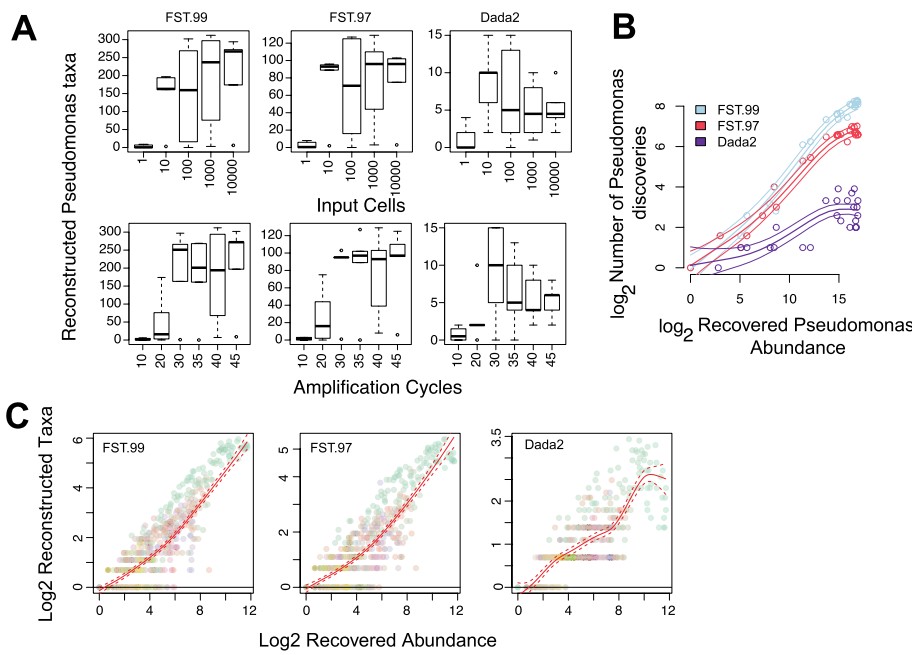

**Fig. 3** False microbial discoveries accumulate along the recovered abundance axis in the Pseudomonas dilution study. **A** For each taxa clustering method, the observed variation in within-genus Pseudomonas taxa accumulations are driven by experimental and technical parameters. Contaminant Pseudomonas are expected to fall with input loads, indicating false discovery accumulations at higher recovered Pseudomonas abundances. **B** The genus recovered abundance axis offers a succinct representation for taxa accumulations. Average and the 95% point-wise confidence intervals for the logged within-Pseudomonas taxa accumulation trends are shown with colored lines for each method, with colored circles indicating the respective observations. **C** An overlay of taxa accumulations across multiple detected genera in the study. Colors indicate genera

those Pseudomonas taxa that track input cells does not change the aforementioned conclusions qualitatively (Additional File 2: Fig. S5).

Similar results on taxa accumulation patterns were also obtained for the *multiple-genera* Oral and Gut mock communities of the microbiome quality control project, handling lab B (MBQC [27]).

Because true taxa are expected to replicate across study samples, we explored sub-genus taxa occurrence rates (Additional File 2: Fig. S19). In all studies, we find that over 50% of sub-genus taxa in over 50% of the detected genera did not replicate in more than 10% of the samples. Mock experimental communities are expected to represent a greater degree of homogeneity than real world communities as the latter may contain rare variants. Restricting analysis to experimental communities with single- and multiple- mock genera, we find that in eight out of nine datasets, over 50% of sub-genus taxa in over 50% of the mock genera replicated in less than 50% of the samples (Additional File 2: Fig. S19). These results indicate poor within-study replicability of most sub-genus taxa.

Finally, because we expect true taxa richness and evenness to vary along the taxonomic tree, we explored taxa accumulations for the various taxonomic levels (i.e., family, order, class and phylum) in each study. Remarkably, the total number of

observed taxa at any level of the taxonomic tree was strongly predicted by recovered abundance alone and was not dependent on the taxonomic level considered (Fig. 2A, S20). These results indicate a strong regularity in taxa accumulations across taxonomic levels, suggesting that spurious taxa accumulations overwhelm observed richness statistics.

Taken together, our results indicate that most sub-genus taxa in 16S surveys are spurious.

### Spurious taxa confound differential richness inference

That observed spurious sub-genus taxa increasingly accumulate with genus recovered abundances leaves us with two expectations.

First, without appropriate corrections, inferring differences in a genus' number of associated taxa (i.e., genus-wise differential richness) are highly likely to be confounded by the genus's respective difference in the recovered abundances (differential abundance). We observe that estimated genus-wise richness values from asymptotic estimators grew systematically with the genus-specific recovered abundances (Additional File 2: Fig. S6). In addition to observed richness, estimates of unobserved richness can exhibit similar behavior (Additional File 2: Fig. S7). This in turn induces an artifactual positive correlation between the resulting genus-wise differential richness fold changes and the genus-wise differential abundance fold changes (Fig. 2B, Additional File 2: Fig. S8).

Second, inferring differential richness between sample-groups (i.e., sample-wide differential richness) are highly likely to be confounded by a net non-zero relative abundance fold change of detected genera. Straightforward simulations where spurious taxa are generated in an abundance dependent fashion illustrate this behavior (Additional File 2: Fig. S6). Interestingly, illustrative examples of the same were rare in several 16S surveys, suggesting that spurious taxa accumulations are comparable at the sample-level. Indeed, in many datasets, the relative abundance log fold changes of member genera were symmetric and concentrated around zero (Additional File 2: Fig. S9-S10). Nevertheless, exceptions with asymmetric relative abundance log fold change distributions exist and a case in point is offered by the long-term time series study discussed below (Additional File 2: Fig. S11).

In Additional File 4, we model the abundance dependent generation of spurious taxa in 16S surveys within the sampling theoretic framework of Chao [29] and Harris [50], and find that the above observations agree with theory.

### Prokounter enables flexible differential richness inference

To overcome the aforementioned biases when applying current richness estimators to 16S surveys and to establish a flexible differential richness inference approach, we developed *Prokounter*.

While zero-truncated statistical models offer one route to modeling member inclusions in a population survey, the same can be achieved by incorporating appropriate predictors in a regression context [51]. The former is the approach taken by some classical richness estimators to model species abundance [28, 52]. We take the latter view and proceed as follows. Based on the results from the "Most sub-genus taxa in 16S surveys are likely technical artifacts" section, we assume that most sub-genus taxa in 16S surveys

are false. This allows us to exploit a 16S survey's overall sub-genus taxa accumulation trend, along with any systematic genus-specific effects, as a sampling effort dependent control for false taxa accumulation (Methods). This control is exploited within standard regression methods for differential richness inference.

With a few 16S surveys, we illustrate the insights offered by the proposed procedure, in achieving genus-specific and sample-wide differential richness inferences.

Unlike other estimators analyzed here (Chao1 [29], ACE [53], Breakaway [54], Breakaway_nof1 [36] and Chiu-Chao [35]), the uncorrelatedness of Prokounter's richness statistics with genus-wide differential abundance statistics is clear in each dataset (Fig. 2C, S12, compare to Fig. 2B, S8). Breakaway and Breakaway_nof1 estimates were the most variable, often accompanied by wide confidence intervals. On several occasions, genus-specific differential richness estimates were not well defined in numerical value when using current richness estimators for numerical and not necessarily statistical identifiability reasons. Sample-wide inferences agreed among all methods in most cases, except when detected genera exhibited a net non-zero relative abundance fold change distribution.

Richness estimates produced by sequencing noise aware estimators Breakaway_nof1 and Chiu- Chao were well correlated with their basic counterparts Breakaway and Chao1 (Additional File 2: Fig. S21), implying that the confounded inferences observed with the latter are likely to be inherited. This was confirmed by simulations (Additional File 2: Fig. S6C). Visualizations of the behavior of Breakaway_nof1 and Chiu-Chao estimators (such as in Fig. 2B) for genus-level inferences were not possible as both estimators were either mostly numerically undefined at lower levels of the taxonomy and/or suffered failed numerical convergence (Additional File 2: Fig. S22). For example, in the time series study discussed below, restricting to those samples where the Chao1 estimates were well-defined, the Chiu-Chao estimator was undefined for over 95% of the time. Because both Breakaway_nof1 and Chiu-Chao methods attempt to correct spurious singleton frequencies based on ratios of higher order frequencies (e.g., doubletons, tripletons), estimation issues arise whenever these ratios are not well-behaved. The above results indicate that these situations are not rare in 16S microbiome data.

In all surveys below, asymptotic genus-wise and sample-wide richness estimates heavily tracked their respective observed richness values (97–100% Pearson correlations, Additional file 2: Figs. S13-S17).

### *Pseudomonas dilution study*

The Pseudomonas dilution experiment varied two parameters of a 16S experimental pipeline: amplification cycles and input cells of a single colony-derived microbial population.

Increased amplification cycles can allow increased sampling of both contaminant and input genera. Thus, within further sampling constraints imposed by the multiplexed nature of the experiment, we expect sample-wide richness to grow with amplification cycles. Sample-wide differential richness inference from all methods matched this expectation.

It is well known that the abundance of lab contaminants falls with input loads [34]. If the dynamic range in input loads is sufficiently high, we can expect inferred sample-wide richness to fall with input Pseudomonas cells. Results from Prokounter, Chao1/Betta [26] and ACE/Betta, Breakaway_nof1/Betta, and Chiu-Chao/Betta matched this expectation. Breakaway/Betta failed to reject the corresponding null hypotheses ($p = 0.4$).

The genus of principal interest in this experiment is *Pseudomonas*. The genus-wide differential richness results from Prokounter indicated a decrease in richness with respect to input cells and an increase with respect to amplification cycles. This is in line with our expectations as we expect the detection rate of lab contaminant Pseudomonas species to grow with amplification cycles and fall with input Pseudomonas loads. In direct contrast, Chao1/Betta, ACE/Betta, Breakaway_nof1/Betta, and Chiu-Chao/Betta, confounded by input Pseudomonas's increasing abundance, indicated a Pseudomonas richness *increase* with input cells ($p = 0$ for both) and Breakaway/Betta failed to reject ($p = 0.251$).

### Long-term time series study

Based on a clustering analysis of abundance profiles, David et al., [2] identified that a distinct sub-group of the phyla *Firmicutes* replaced another *Firmicutes* sub-group, post-enteric infection, in the gut microbiome of an individual relocating to a different country. Prokounter refines this result further by identifying several Firmicutes genera (*Faecalibacterium*, *[Ruminococcus]*, *Oscillospora*) that are less rich post-infection. On the other hand, *Dorea* and *Coprobacillus*, members of Firmicutes, were found to have significantly increased richness in infection and post-infection samples. The genus *Acinetobacter* from the phylum *Tenericutes* was found to have significantly increased richness in samples collected during infection, while this was not the case post-infection. Thus, differential richness adds another state variable to the microbiome state specifications of the original study.

In David et al.'s dataset, sample-wide inferences disagreed among the methods compared. Relative to pre-infection samples, Prokounter generated inferences of reduced richness for both infection and post-infection samples consistent with antibiotic exposure. Chao1/Betta and ACE/Betta indicated reduced richness post-infection with a relatively weak significance for reduced richness in infection samples. Breakaway_nof1/Betta and Chiu-Chao/Betta indicated reduced richness post-infection but failed to reject the null for infection samples ($p$-value$= .10$ Breakaway_nof1/Betta, $p$-value$= .16$ Chiu-Chao/Betta). Breakaway/Betta failed to reject any of the corresponding null hypotheses ($p$-value$= 0.99$ infection and $p$-value$= 0.74$ post-infection), potentially owing to the very high variability of Breakaway estimates. As established in the previous subsection, these differences in inferences likely stem from the asymmetric differential abundance of detected genera in the samples collected during infection.

### Waste-water treatment

To demonstrate an ecological monitoring application, we applied Prokounter to 16S data arising from a waste-water treatment plant [55]. The method indicates that relative to the effluent, sample groups from each of the post-treatment stages have significantly reduced microbial richness values. These results readily agree with our expectation of

a publicly implemented waste-water treatment protocol. Chao1/Betta, Chiu-Chao1/Betta, and ACE/Betta produced similar results. While Breakaway_nof1/Betta failed to reject the null for inlet to pumphouse ($p = 0.169$), Breakaway/Betta failed to reject the null for sample groups corresponding to both effluent ($p = .065$) and inlet to pumphouse ($p = .692$).

Using differential abundance analysis, the original study highlighted the persistence of *Legionella* and *Mycobacterium* in post-treatment samples calling into question the efficacy of the treatment process. Performing genus-specific differential richness analysis with Prokounter indicates that the treatment plant reduces the richness associated with several types including *Mycobacterium*. We did not detect *Legionella* as reduced in richness in the effluent. These results indicate that waste-water treatment has been effective with removing *Mycobacterium* sub-types.

### *Pathogenesis*

We applied Prokounter to a 16S survey of the cerebrospinal fluid from hydrocephalus children hypothesized to have infectious (PIH) and non-infectious (NPIH) origins [56]. We intuitively expected, and observed, that the cerebrospinal fluid enveloping the central nervous system to register lower richness compared to laboratory controls. PIH samples had relatively lower richness compared to clinical control samples.

A genus that is more abundant with reduced richness might indicate invasion of a sub-species. Genus-specific differential richness inference with Prokounter yields two genera as having lowered richness in the PIH samples: *Paenibacillus* and *Streptococcus*. *Paenibacillus* was the dominant pathogenic genus identified with the PIH phenotype using 16S data [56].

## Discussion

### Summary

16S microbiome surveys reconstruct target microbial populations by clustering sequencing reads. Spurious microbial taxa occur when the clustering procedure's error model fails to capture the entirety of sequence variation induced by the technical steps in 16S sequencing (Additional File 1, Fig. 1A). We have shown that the false taxa thus generated not only inflates the estimate of a (microbial) community's richness (Additional File 4, Additional File 2: Fig. S6), but they also cause taxa differential abundance to confound differential richness inferences (Fig. 2B, Additional File 2: S8). This occurs because every false taxon is generated through errors from one or a few true (i.e., input) taxa, and hence, their rates of production increase with the output abundance of the corresponding source taxa (Additional File 1). Based on our result that most sub-genus discoveries are likely false (the "Most sub-genus taxa in 16S surveys are likely technical artifacts" section), we have established abundance dependent controls for false taxa accumulations using a given survey's within-genus taxa accumulation data (Methods, Fig.2C, S2, S18). We have shown that our strategy overcomes the confounding problem (Fig. 2C, S12). And we have illustrated the utility of differential richness inferences in individual and public health-related microbiome data analyses (the "Prokounter enables flexible differential richness inference" section).

**Assumption**

Our approach assumes that most sub-genus taxa in 16S surveys are spurious and are poor representatives of the underlying microbial community. We have provided several lines of evidence to support this conclusion: First, a mock experiment of an overnight derived microbial population indicated that observed richness can be severely inflated (Fig. 3, S5). Our expectation was set in part by a mathematical model of cellular reproduction, where we tracked the probability distribution over substitutions, over generations (Additional File 1). Second, in a manner similar to what we would expect of low probability errors, most sub-genus taxa in both controlled mock and real world datasets are rare and show poor replicability across samples (Additional File 2: Fig. S19). Third, the total number of taxa observed for any taxonomic level was strongly determined by the category's recovered abundance alone and was not dependent on the level itself (Fig. 2, S20). Fourth, within-genus taxa accumulation patterns in several publicly available datasets, including those from single- and multi-genera mock experiments, appear remarkably regular as if most genera in 16S surveys have similar richness and taxa evenness (Figs. 2 and 3, S1-S3, Tables 1 and 2, Additional File 3: Tables S2-S3). Finally, the literature offers abundant support for abundance dependent false taxa generation in 16S surveys, of which we note a closely related few. Kunin et al., [32] demonstrate the large number of false *Escherichia* taxa that arise in a 16S survey of a target *E. coli* population (also see Degnan and Ochman [57], Pinto and Raskin [58]). Based on the empirical observation that the number of false taxa generated are sampling effort dependent, Schloss et al. [59] recommend that community-level comparisons be made at comparable sampling depths. Haas et al., [60] illustrate the predictable, abundance-dependent generation of false chimeric taxa within genera in mock communities. Finally, consistent with our results on spurious taxa accumulations, Fouladi et al. [61] demonstrate the abundance dependent accumulation of 16S sequencing read error variants in microbiome datasets.

**Implications for richness theory and automated ecological surveys**

False microbial taxa in 16S surveys arise because automated procedures to reconstruct taxa misclassify sequencing reads from their true types. Thus, in Additional File 1, we analyzed the influence of amplification and sequencing induced substitutions in causing misclassifications (also see Schloss [62] and Sze and Schloss [63]). In Additional File 4, we mathematically modeled the false taxa that arise through misclassification and showed in part that a traditional asymptotic richness estimator (Chao1 [29]) is biased under this more general sampling scenario. The severity of bias is determined by sampling parameters. Together with the results mentioned in the previous paragraphs, we conclude that classical richness theory, which predominantly focuses on estimating undetected richness while assuming observed richness at face value, should be generalized for observed species misclassifications in modern high throughput and highly automated surveys.

**Asymptotic richness estimators track observed richness values in 16S surveys**

In the several 16S surveys considered here, asymptotic richness estimates tracked observed richness values both sample-wide and at within-genera levels (Additional File

2: Fig. S13-S17). Our mathematical models and simulations that incorporate false taxa accumulations within the sampling theoretic framework of Chao [29] and Harris [50] indicate that such tracking can arise when the apparent richness (i.e., the true plus false richness) and not necessarily true richness is undersampled in a survey (Additional File 4). This explains the observed tracking in the *Pseudomonas* genus in the *Pseudomonas* dilution experiment, where we do not expect undersampling of the true *Pseudomonas* community (Additional File 2: Fig. S13).

### False discovery control in differential richness analysis, confounding with differential abundance

Hughes et al. [64] argue that traditional macroecological richness estimators continue to enable robust sample-wide richness comparisons in 16S surveys. Our analysis identifies exceptions (the "Long-term time series study" section) and clarifies the practical conditions under which controlling for spurious discoveries become important. In particular, we find that false taxa accumulations cause abundance dependent inflation in observed taxa numbers and their frequencies (Additional File 1 and 2), causing differential (relative) abundances of detected taxa to confound differential richness inference with traditional methods (Fig. 2B, Additional File 2: S6-S8, S11). When spurious taxa accumulations are comparable across contrasted experimental groups, no such confounding arises (Additional File 2: Fig. S9-S10). Our empirical analyses indicate that such an assumption is too strong for making differential richness inferences at lower taxonomic levels (e.g., genus-specific) of a microbial assemblage (Fig. 2B). Unlike human gene expression studies that enjoy a known nucleotide reference catalog to reconstruct gene expression, microbiome studies often lack a well-defined reference. Thus, microbiome studies typically reconstruct the target microbial population bioinformatically by clustering 16S sequencing reads for sequencing noise, with error models derived based on sequence similarity thresholds, or by modeling sequencing noise. However, arriving at accurate error models is hard in practice (e.g., higher order sequence dependence of errors is not modeled), and thus spurious microbial discoveries are bound to occur when typing several millions of sequencing reads (Additional Files 1 and 2). It is in this context that we find utility with our proposed differential richness inference method, which aims to estimate and correct for the generation of spurious taxa in highly automated microecological surveys.

### Relaxing microbiome richness comparisons to taxonomic groups

Microbiome analyses frequently restrict richness comparisons to the entire microbial assemblage obtained in study samples (sample-wide richness inference). From the perspective of deriving health and ecological indicators based on community assemblages, analysis of a community's finer organization levels is equally interesting [2, 8, 10–12, 17]. Our genus-wise differential richness results (the "Prokounter enables flexible differential richness inference" section) indicate that contrasting richness for taxonomic sub-groups can enable practically useful inferences and add interesting dimensions to microbiome state space descriptions.

### Within-genus taxa accumulation structure and the trend estimator

Our results document reliable across-genera regularity in the patterns of within-genus taxa accumulations, across many studies and genus-specific experiments (Figs. 2 and 3, S1-S3, Tables 1 and 2, Additional File 3: Tables S2-S3). We speculate that genus abundances, in contrast to sampling depth, more accurately track the sampling rate of false sequence variation in 16S surveys for at least two reasons. First, commonly exploited 16S rRNA target segments are limited in resolution beyond genus level [42–49]. Second, genus recovered abundances, unlike total sampling depth, normalize for the sampling rates of distinct genera. This restricts us from mixing taxa accumulation statistics over truly disparate input biological sequences from distinct genera, while allowing us to preserve any systematic genus specific effects. We used a robust trend estimate of the within-genus taxa accumulation data to model spurious taxa accumulation (Methods, Figs. 2 and 3, S1-S3). The coherent accumulation of a large number of detected taxa translated to low estimation uncertainties. These curves were not necessarily linear in the recovered genus abundances (Additional File 2: Fig. S1-S3). The systematic genus-specific contributions to this trend can arise due to between-genera variation in both detectable true input sequence diversity (copy number [48] or number of distinct cell types) and 16S sequencing noise [62, 63].

### Emphasis on sub-genus taxa

We have exploited the characteristics of sub-genus taxa accumulations to estimate controls for the generation of spurious taxa in a microbiome survey. However, our results indicate that estimation of the needed controls could also be done based on higher order taxonomic groupings as they too exhibit similar taxa accumulation characteristics (Fig. 1). In principle, we may lose power in resolving genus-specific technical effects (as illustrated in Additional File 2: Fig. S18), and this necessitates a discussion of two cases. First, if grouping of genera is made because of very similar/identical 16S sequences (e.g., Escherichia-Shigella), we may not need to account for individual genus-specific effects within the grouping, as sequencing errors are strongly dependent on nucleotide sequence context. Second, when taxonomic descriptions for sequences are incomplete or not available, based on the comprehensive work of Yarza et al. on rational taxonomic boundaries [47], we recommend a numerical measure of 94.5% sequence identity for typing arbitrary collections of 16S sequences to achieve a reasonable "Genus-level" grouping for estimation purposes.

### Abundance dependent control in bioinformatic sequence analysis

Beyond differential richness inference, there is a need for recovered abundance dependent control in other (meta)genomic sequence analyses, e.g., sequencing read mapping and taxonomic annotation, which exploit fixed sequence similarity thresholds. Probabilistic methods have a natural incorporation of abundance in clustering/mapping decisions. In all cases, however, poor error models would continue to drive false taxa accumulations. It must be noted that we have not analyzed false negative rates in this study [65, 66].

### Limitations of differential richness inference

Observed (and reportedly, asymptotic [67]) richness estimates cannot forecast crossing over of species accumulation curves that can in principle occur with additional sampling effort. However, differential analysis of both these estimates over realized sampling effort is still useful for detecting perturbations to the evenness of a biological community [64, 68] and is thus effective for deriving predictors of individual and environmental health.

### Future work

There are several avenues for future research. First, an integrated estimation procedure of false taxa accumulation rates and differential richness fold changes would lead to more appropriate *p*-values under the assumed statistical models. Second, development of ecological richness estimators in the presence of species misclassifications would be a valuable addition to the literature. Additional File 4 considers a simple but a useful special case. Third, 16S surveys on mixtures of microbial species with varied relatedness and controlled input richness levels would enable a joint characterization of detectable 16S resolution, taxa reconstruction algorithms, and richness estimators. Fourth, control for multiple testing over tree structured hypotheses can be incorporated if one wishes to automate hypothesis testing over taxa collections defined by subtrees of a taxonomic tree [69, 70]. Finally, all our empirical observations were based on a set of 16S surveys that operate over partial 16S gene targets. Because full length 16S surveys also involve amplification, and sequencing protocols [46], we expect the qualitative nature of our results to generalize to such surveys, perhaps at a lower taxonomic level (e.g., species), and this can be explored.

Taken together, this paper significantly clarifies the dynamics of spurious discovery accumulation in 16S surveys, presents strategies for modeling their generation, demonstrates the need to control for the observed false discoveries in microecological surveys while deriving differential richness inferences, and offers a flexible practical solution to achieve the same.

### Conclusions

Estimates of species richness based on classical richness estimators are biased when ecological surveys include false species generated by misclassifying true species. In the specific case of 16S surveys, the rate of generation of false taxa strongly increases with the output abundances of true input sequences. Thus, changes in the abundance of detected taxa are mistaken for changes in richness by current methods for differential richness inference. The results in this paper lead to the conclusion that most sub-genus taxa in 16S surveys are spurious. Based on this finding, a flexible method for differential richness inference was proposed and was shown to the overcome the confounding problem. The paper clarifies the dynamics of spurious taxa accumulations in 16S microbial surveys and argues for the development of diversity estimators that adapt to species misclassifications in modern, highly automated ecological surveys.

## Methods

### Prokounter

Our proposed procedure for differential richness inference works in two steps. A control for false taxa accumulation is established first. The estimated control is subsequently exploited within standard generalized linear models for differential richness inference.

Let $n_{gj}$ denote the reconstructed number of taxa for genus $g$ in sample $j$, $y_{gj}$ denote the corresponding recovered abundance (i.e., genus's total count in the sample), and $\tau$ represent the sample depth.

Let $f_t(log y_{gj})$ indicate the logged technical contribution to taxa accumulation for a given genus and its recovered abundance level. This function is used to model the log of the expected false taxa accumulation. Its estimate $\hat{f}_t(\log y_{gj})$ is obtained using within-genus taxa accumulation data as follows.

### Estimating the technical contribution $\hat{f}_t$

We explored two strategies to estimate a robust within-genus accumulation trend.

A semi-parametric smoothing spline model is assumed on $z_{gj} = log\, n_{gj}$,

$$z_{gj} \mid g, y_{gj} = \eta(g, y_{gj}) + \varepsilon_{gj} = \kappa + f_R(\log y_{gj}) + f_G(g) + f_{GR}(g, \log y_{gj}) + \varepsilon_{gj} \qquad (1)$$

with $\varepsilon_{gj} \sim N(0, \sigma^2)$, and appropriate side conditions are placed on $f.(\cdot)$ (Chapters 2-3 [71, 72],). Here, $\kappa$ and $f_R(\cdot)$ denote the intercept and recovered abundance dependent components; $f_G$ and $f_{GR}$ indicate the genus and its respective interaction functions with the recovered genus abundance.

Briefly, $\eta$ is estimated as a unique solution to the penalized optimization problem: $\hat{\eta} = \arg min_{h \in H}\, l(h \mid \tilde{y}, x.) + \lambda J(h)$, where $l(\cdot \mid \tilde{y}, x.)$ is the negative log likelihood, $\lambda$ is a regularization parameter, and $J(\cdot)$ is a roughness penalty that penalizes overfitting of $h$ to the data. The specification of $J(\cdot)$ involves, in part, integrals of squared second order derivatives of the estimand over the range of $log y_{gj}$, thereby enforcing smoothness. Additional File 6 offers more details on the model construction and an exact correspondence to example 2.7 in Gu [73]. Numerical optimization is perf ormed using the R package *gss* [71]. Supplementary figures Additional File 2:S2 and Additional File 2: S18 offer examples of the fits that result.

The technical contribution to taxa growth is estimated as $\hat{f}_t(g, \log y_{gj}) = \kappa + \hat{f}_R(\log y_{gj}) + \hat{f}_G(g)$. Only the significant genus effects are retained after multiple testing correction with the Benja mini-Hochberg procedure. When the genera contributions are null or similar, as we observed empirically in several datasets (e.g., Additional File 2: Fig. S9, S10), $\hat{f}_t(g, \log y_{gj}) \propto \hat{f}_R(\log y_{gj})$

The latter observation inspires the following alternative strategy: estimate $\hat{f}_t(\cdot)$ as a net average within-genus accumulation curve using the *loess* smoother. Both options are made available in our software. As expected, inferences arising and the results in Tables 1 and 2 are similar with both approaches. Additional File 2: Fig. S3 offers examples of the fitted trends. The spline strategy does offer better control in the presence of systematic genus effects (Additional File 2: Fig. S18).

For consistency, in this paper, we have chosen the spline strategy.

The fitted $\hat{f}_t(\cdot)$ can be used to control for false taxa accumulation in standard differential richness inference procedures. In *Prokounter*, we incorporate it through the models presented below.

**Differential richness in ference**

We use Greek letters to indicate regression parameters. A ·in the subscript indicates vectorizing over the subscript. $X$ denotes the experimental design matrix. Genus-specific, sample-wide, and taxa collection models are presented in Equations (2–4) below. In each case, given the quantity modeled, reasonable transformations of the estimated logged technical contribution, $\hat{f}_t(\cdot)$, based on Eq. 1, are used. Terms involving X below can be viewed to approximate the effects arising from genus-recovered abundance interaction terms from eqn. 1.

### *Genus-specific differential richness inference*

The conditional mean of the observed richness is modeled through the link:

$$logE\left[n_{gj}|\, y_{g\cdot}, X, f_t(\cdot)\right] = X_j^T \mu_g + v_g f_t\left(log y_{gj}\right) \tag{2}$$

where the right hand side is an approximate form for the log of the conditional expectation of the right hand side of Eq. (1).

### *Sample-wide differential richness inference*

For inference across sample groups, we posit:

$$logE\left[n_{+j}|\, y_{g\cdot}, X, f_t(\cdot)\right] = X_j^T \zeta + \gamma \, log\sum\nolimits_{g:y_{gj}>0} e^{f_t(log y_{gj})} \tag{3}$$

where the $+$ indicates summation over a subscript. As in Eq. (2), the right hand side of Eq. (3) is an approximate form for the log of the conditional expectation of the right hand side of Eq. (1), now summed over *g*. The net sample-wise technical contribution is modeled as a simple sum of the technical contributions from the genera detected in the sample. Although Eq. (3) does not immediately arise from eqn.(2), we find the simplicity and emphasis on dominant contributors to the sum, the more abundant genera, appealing. In addition, we often find that $v_g \approx 1$ and $\gamma \approx 1$ in applications.

### *Differential richness inference for arbitrary collections of genera*

For an arbitrary taxonomic group *k* (e.g., phyla), with a set of member genera $G_k$, we assume:

$$logE\left[n_{kj}|\, y_{g\cdot}, X, f_t(\cdot)\right] = X_j^T \psi_k + \gamma_k log\sum\nolimits_{g \in G_k \cap y_{gj}>0} e^{f_t(log y_{gj})} \tag{4}$$

As with the sample-wide model, here too, we have modeled the sample-wise technical contribution for each collection *k* based on the sum of genus-level technical contributions, but now restricted only to those genera considered within the collection.

Keeping to the traditional theme of continuous Poisson mixtures driving sample-wide species accumulations, we chose negative binomial variance functions when performing sample-wide inferences and Poisson variance functions for genus-specific richness inferences. For the several studies considered here, the estimated overdispersion coefficients for sample-wide negative binomial models were in the range of $10^{-3}$ to $10^{-1}$. For well expressed genera, inferences and model diagnostics were not sensitive to the two

distribution assumptions. Parameter estimation and inference on the regression parameters $\mu_g$, $\zeta$ and $\psi_k$ were performed using R's *glm* function. Maximum likelihood estimation with iteratively reweighted least squares converges rapidly in about ten iterations or less. Speaking to the explanatory power of $\hat{f}_t$, as implied by Tables 1 and 2, the residual deviance is often small, on the order of the residual degrees of freedom. To gauge reproducibility of inferences over fitted $\hat{f}_t(\cdot)$, confidence intervals based on the bootstrap $t$ [74] are also available for the regression coefficients of the sample-wide differential richness inference model.

The above models, which were used to generate the results in the applications section, exploit observed richness as response variables and are therefore non-asymptotic in nature. In the several 16S surveys considered here, asymptotic genus-wise and sample-wide richness estimates heavily tracked their respective observed richness values (97–100% Pearson correlations, Figs. S13-S17). We therefore propose the same regression models above for standard inverse variance weighted regression analyses of asymptotic richness estimates. As expected, results from such a procedure were similar to those obtained with observed richness as the response variable. Also see ref. [26] for a heterogeneity test of potential interest.

We implement these procedures in an R package *Prokounter*. Additional File 4 presents further discussions on the regression models above.

### Package and code availability

The R package *Prokounter* is available at the URL: https://github.com/mskb01/prokounter

Code for the paper is available under https://github.com/mskb01/prokounterPaper

### Richness estimators and differential analyses

Estimates and standard errors for Chao1 and ACE estimators were calculated using the R package *vegan* [75]. Breakaway and Breakaway_nof1 estimates and standard errors were obtained using the R package Breakaway. Chiu-Chao method's estimates and standard errors were obtained using the source code attached to the original paper [35]. Differential richness inferences corresponding to the five estimators were obtained with the R package Betta [26]. Rarefaction based interpolated and extrapolated richness estimates and standard errors were obtained using the package *iNext* [76]. The R package doParallel [77] was used for several parallel computing tasks.

The following datasets and study design variables were used to construct design matrices for sample-wide and genus-specific differential analyses reported in the applications section.

1. Hydrocephalus [56] (PIH100 FST97)—control and case
2. Wastewater [55] (WW FST99)—Influent, effluent, before uv treatment, after UV treatment, pond storage, and inlet to pumphouse for subsequent spray irrigation
3. MBQC, Handling lab B (MBQC-HLB) - Gut mock, oral mock, the rest of the stool samples were typed as other

4. Time series study [2] (TS FST97, Donor B)— based on the original study, three time windows were established to define sample groups: days up to 150 were categorized as *pre-infection*, days from 151 u pto 159 as *infection*, and days post 159 were typed as *post-infection*

5. Pseudomonas dilution study (Pseudomonas FST97)—number of cycles and logged number of input Pseudomonas cells

### Dilution experiment

A monoisolate was prepared overnight from a Luria-Bertani (LB) agar plate into a 5 mL LB liquid, which grew to $10^9$ cells. A tenfold serial dilution of cells from $10^5$ to 10 cells in phosphate buffer saline (PBS) was generated. DNA was isolated, 16S amplified, and sequencing libraries were prepared as previously described [56]. Briefly, DNA was isolated using the Zymobiomics DNA miniprep kit following manufacturers' protocol with bead beating and proteinase K treatment. For 16S amplification, primer-extension polymerase chain reaction (PE-PCR) of the V1-V2 region was performed using an M13 tagged 336R universal primer as previously described [78], and amplification cycles were varied. Briefly, target DNA was mixed with a 10 µl of 10X buffer and annealed with M13 tagged 336R by first heating to 95 °C and then cooling to 40 °C slowly. The annealed product was extended using Klenow polymerase (5 U/µl and primers digested with 20 U/µl Exo I (NEB, USA), then amplified with 500 nM primers (805R and M13) using the MolTaq 16S Mastermix (Molzym GmbH & Co Kg, Germany). Library preparation was done with the Hyper Prep Kit (KAPA Biosystems, USA) following the manufacturer's protocol and libraries were sequenced on MiSeq using the 600 cycle v3 kit. Sequencing reads are available from SRA under the Bioproject ID: PRJNA779422.

### 16S datasets and taxa reconstruction pipelines

All datasets were obtained as described under the "Availability of data and materials" section. We generated three varieties of taxa count data from each of the *Pseudomonas*, *PIH100* 16S, and MBQC *HL-B (handling lab B)* sequencing data. These include sequence similarity threshold based taxa clustering methods for 99% and 97% sequence similarities (*Qiime1*), and a probabilistic taxa clustering method (*Dada2*) as follows.

### Quality filtering of sequencing reads

Paired-end reads were processed with *Trimmomatic* [79] (v0.38) to remove universal adapters and low-quality reads. Reads with ambiguous bases were removed or truncated using *Dada2*'s *filterAndTrim* [80] function. The 16S V1-V2 regions in both our Pseudomonas and PIH100 data were sequenced using 2x300bp paired-end reads. Based on sequencing read quality score profiles, we retained the first 240 bp and 210 bp in the forward and reverse reads for the *Pseudomonas* dataset. These numbers were respectively 200 bp and 190 bp for *PIH100*. For HL-B, we removed the first 2 bp following the primers in the forward and reverse reads. This allowed us to neglect the trailing low quality bases

adversely affecting the taxa reconstructions, while still allowing for sufficient overlap to merge paired-end reads.

Reads with either the designed primers or their reverse complements were filtered using *cutadapt* [81]. The quality filtered reads were then clustered with Qiime1 [82] and Dada 2[80] as below.

### Qiime1

Quality filtered forward and reverse reads were merged using *Pear* [83], and then clustered using *pick_open_reference_otus.py (Qiime1 version 1.9.1),* which implements the Qiime1 open reference OTU clustering algorithm. Briefly, closed reference clustering of merged reads were performed against the *Silva132* database at 97% and 99% sequence similarity thresholds, using Uclust [84] v.1.2.22q . Reads that did not map to the database were subsampled and used as new centroids for a de novo OTU clustering step at the respective sequence similarity thresholds. Remaining unmapped reads were subsequently close clustered against the de novo OTUs. Finally, another step of de novo clustering was performed on the remaining unmapped reads. Taxonomy was assigned to taxa representative sequences with Uclust based on the *Silva132* [85] database. These sequences were filtered with *Pynast* [86], and OTU tables generated.

### Dada2

Dada2 allows denoising forward and the reverse reads independently. Error rates were estimated separately for the quality filtered forward and reverse reads for each sample. This estimation step is based on a sample of reads for computational tractability. Reads were deduplicated and sequence clusters inferred based on the estimated error rates. Taxa from forward and reverse reads were merged at the end of the workflow. Chimeric taxa were removed with the function *removeChimeraDenovo*. The resulting taxa were assigned taxonomic labels based on the *Silva132* database, using their naïve Bayes classifier.

### Supplementary Information

Additional file 1. Presents the theoretic models for substitution errors introduced during amplification and sequencing, incorporating back mutations. The influence of recovered abundance-dependent accumulation of false sequencing read classifications is characterized. Related figures and literature references are contained within the note.

Additional file 2. Presents supplementary figures S1-S22. Related literature references are contained within the note.

Additional file 3. Presents supplementary tables S1-S3. Related literature references are contained within the note.

Additional file 4. Presents a theoretic model for the generation of false species by misclassifying true species in a population survey. Characterizes the influence of false species accumulation on the observed and estimated unobserved richness components underlying standard richness estimators. Related figures and literature references are contained within the note.

Additional file 5. Presents simulation details for studying the confounding introduced by between-sample differential abundance of detected taxa in between-sample differential richness.

Additional file 6. Presents details on smoothing spline model construction. Related literature references are contained within the note.

Additional file 7. Review history.

**Acknowledgements**
This project was supported by an NIH Director's Transformative Award 1R01AI145057. We thank the Genome Science Facility at the Penn State University College of Medicine for performing the sequencing for the *Pseudomonas* dilution study.

**Peer review information**

**Review history**
The review history is available as Additional file 7.

**Authors' contributions**
Conceived and designed study: MSK, SJS, JNP. Theory: MSK, EVS. Collection of publicly available data: MSK, LZ, JNP. 16S sequence processing pipeline design: LZ, MSK. Execution: LZ. Pseudomonas experiment's design: CH, MSK, JB. Execution: CH. Data analysis: MSK. Software: MSK. Interpretation of results: All authors. Wrote the paper: All authors. The authors read and approved the final manuscript.

**Funding**
This project was supported by an NIH Director's Transformative Award 1R01AI145057.

**Availability of data and materials**
All data generated or analyzed during this study are included in this published article and its supplementary information files. The mouse microbiome data was obtained from the R/Bioconductor package *metagenomeSeq* [87, 92]. The moderate to severe diarrheal data was obtained from the R/Bioconductor package *msd16S* [88]. The long-term time series data [2] was obtained from the supplementary data of the corresponding paper. The wastewater 16S survey [55] was obtained on request from the authors of the original study. MBQC handling laboratory B's (HL-B) sequencing reads was obtained from the Microbiome Quality Control (MBQC) project [27]. S equencing reads for the Pseudomonas dilution study are available from SRA via the Bioproject ID: PRJNA779422 [89].
The R package *Prokounter* is available at the URL: https://github.com/mskb01/prokounter, and released as https://doi.org/10.5281/zenodo.6654769 [90]
Code for the paper is available in: https://github.com/mskb01/prokounterPaper, and released as https://doi.org/10.5281/zenodo.6654767 [91]

## Declarations

**Ethics approval and consent to participate**
Not applicable.

**Consent for publication**
Not applicable.

**Competing interests**
The authors declare that they have no competing interests. JNP is an employee of Genentech, Inc.

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

## 
