## [Additional file 7. Review history. · Genome Biology]

Review History

First round of review

Reviewer 1

Are you able to assess all statistics in the manuscript, including the appropriateness of statistical tests used? Yes, and I have assessed the statistics in my report.

Comments to author:

In this manuscript, Kumar and colleagues argue that nearly all taxa observed at the sub-genus level from de-novo OTU clustering or sequence variant detection algorithms are results of sequencing error and that the relationship between taxa abundance and the number of variants observed within that taxa can be easily modeled to produce better descriptions of richness than naïve algorithms that do not take into consideration sequencing error. The authors imply that algorithms like DADA 2 are not working very well in their attempts to correct for sequencing based error rates and that Qiime based OTU clustering at 99% or 97% do not guard against sequencing error causing spurious OTUs. They propose models that explicitly consider rates of sequencing error as correctives. The paper is quite clearly written and both compelling and convincing. It makes an important contribution to our understanding of how classical ecological metrics of species diversity interact with modern sequencing technology.

In some ways, the paper refutes early papers in next-generation sequencing metagenomics such as the PNAS paper in 2006 (<https://www.pnas.org/content/103/32/12115/>; "Microbial diversity in the deep sea and the underexplored "rare biosphere"") that argued that there was a "rare biosphere" that represented "a nearly inexhaustible source of genomic innovation". Through a modern understanding such as the one the authors present, it seems we can better understand the observed diversity in such early works as accumulation of sequencing error governed by simple Poisson processes and not related to underlying biology. The data presented in Fig. 2 in the manuscript under review are very compelling in demonstrating beyond doubt that observed richness is a simple and essentially strict function of abundance of the total genus relative abundance. These observations also seem highly consistent with recent observations that the number of variants in 16S datasets can be very well modeled by Poisson sampling taking into account the abundance of each separate variant (as seen in Fig. 2 here:

<https://journals.asm.org/doi/full/10.1128/mSystems.00697-21> HashSeq: a Simple, Scalable, and Conservative De Novo Variant Caller for 16S rRNA Gene Data Sets). The authors may wish to consider citing this recent paper (full disclosure, your reviewer here is Anthony Fodor, last author of that paper).

Overall, the paper is meticulous in its execution and arguments and a model of clarity in presenting complex ideas. If the paper has a limitation, it is that the description of the application of Prokounter to previous datasets could perhaps be a bit clearer and more compelling. In line 238, for example, the authors write that "Prokounter produced negative richness inferences for both infection and post-infection samples consistent with antibiotic exposure." But does "negative" here mean a negative result or a direction of correlation that is negative? This section could be made more clear perhaps by showing a figure with a richness difference analyzed

"traditionally" and with Prokounter. Likewise, similar figures could be made for the Pathogenesis, Wastewater treatment and Pseudomonas datasets. A visual demonstration of the impact of the author's approach on "real world" inference problems in comparison to classical, but naive, approaches might make the paper more accessible for less mathematical readers.

Reviewer 2

Are you able to assess all statistics in the manuscript, including the appropriateness of statistical tests used? Yes, and I have assessed the statistics in my report.

Comments to author:

In microbiome studies, the total number of species present (richness) is a key ecological parameter for microbial communities, but is notoriously difficult to measure accurately. The number may be underestimated if not all species are sampled, but more often is overestimated due to errors in PCR amplification and sequencing. The authors of this paper present a new method to estimate richness by breaking the problem down and estimating the richness of defined bacterial groups, rather than the full ensemble. With this basic idea, the authors cleverly recognize that the number of spurious species introduced from sequencing error will increase with the relative abundance of any real species present.

The method is original, clever, important, and will be valuable to researchers in the field. However, the manuscript is hampered by two major conceptual issues that prevent it from presenting a clear and solid argument for the approach.

1. "Sub-genus taxa."

The authors have placed an enormous amount of emphasis on the taxonomic rank of genus as their primary organizing structure for bacteria. This emphasis is misplaced. The generation of spurious 16S sequences is a purely numeric phenomenon that has to do with PCR errors and fidelity of sequencing instruments. In contrast, a genus is a taxonomic construct invented to name organisms, which is determined using phylogenetic arguments rather than sequence similarity. It just so happens that the typical size of a genus in sequence space coincides approximately with a convenient boundary beyond which spurious species are unlikely to be generated by technical error. The authors might be able to "get away with" using bacterial genera to fuel their software, but I would argue that it is conceptually inappropriate and a gigantic missed opportunity.

The idea of a bacterial genus is fundamentally mismatched with the idea of a group of similar 16S sequences that are generated by technical errors. New genera (in modern times) are established using phylogenetic arguments, which are mismatched with the decidedly non-phylogenetic nature of PCR errors. Genera are imperfect--*Escherichia/Shigella* is the classic example, but the examples of *Eubacterium* and *Clostridium* should scare the heck out of the authors. Genera are incomplete--should the lack of well-established genera for mouse intestinal bacteria (*S24-7/Muribaculum*) and segmented filamentous bacteria (SFB) hold this tool back from accurate estimates of richness?

The method presented here should be applied to sequence groups that are determined numerically, in a way that matches the process by which sequencing errors are generated. The authors have already presented the solution in their paper, when they explore "taxa accumulations at various taxonomic levels" (line 170). This argument confused me at first, but I think it's a key insight from the study. If the sequence grouping is too broad, you still recover the correlation between relative abundance and species richness. So there is nothing special about the genus level! It seems that your groupings must be sufficiently broad to capture the error-generated species along with the true species, but narrow enough that the relative abundance can be used to back out the numbers you need.

I do want to recognize that using the language of "genus-level" similarity makes for a catchy and easy-to-understand results section. If you are able to define "genus-level" similarity in some numerical way, I think you would be safe to state the definition and then stick with your current language.

2. Denoising

Any discussion of denoising methods is conspicuously absent from your paper. You use DADA2 to generate sequence variants, and the denoising process used by DADA2 is supposed to solve exactly the problem you're trying to solve, but you say nothing about it. Ideally, I would run my denoising software, it would characterize all the sequencing errors and remove all the spurious variants, then return a table of authentic 16S sequences. I mean, we all know that's not how it works, but you should explain why you are taking a post-denoising regression/optimization approach, rather than trying to write a denoiser yourself.

One might argue (not me, but someone else) that your problem should be solved by the denoiser. Why is it not solved by current denoisers? Is it best to even approach the problem in this way? What is the relationship between your software and denoising software?

Minor issues:

1. Straw man richness estimators. As soon as you establish that unique sequences are generated by error, all the assumptions justifying the Chao1 estimator fly out the window. It is obviously not going to work, so why are you using it in your comparisons? You should compare instead to your real competitors, like Chiu & Chao and most definitely Willis. Chao1 is pretty lame here.
2. When you write ACE/Betta (line 239), I don't know what the Betta means.
3. "Taken together, our results indicate that most sub-genus taxa in 16S surveys are likely spurious." I think you're being too guarded here: you pretty much know that they are spurious, and you have a pretty good idea about what the exact number is.
4. The *Pseudomonas* dilution study is a much cleaner and simpler example, suggest covering this before the other examples in 2.3.

Software issues:

1. Thanks for making this a real, honest-to-goodness R package. Please finish filling out the DESCRIPTION file.

2. Your dependencies should be in the DESCRIPTION, not sprinkled inside your functions. That way, the dependencies get installed automatically when your software is installed. <https://r-pkgs.org/description.html#description-dependencies>
3. Think carefully about the number of dependencies that go along with your package. MASS and vegan are likely required for basic functionality. Can RColorBrewer and metagenomeSeq be suggested dependencies?
4. In getProkounterTrends(), you have a keyword argument, plt=FALSE, that returns a plot instead of a data object. This is a common anti-pattern in R libraries. Instead, simply return the data object. Pull that plotting logic out into a second function that takes the data object and returns a plot. Not only will everything be easier to use and test, but it will open some space for you to allow the user to configure the plot.

Response to Reviewer #1

In this manuscript, Kumar and colleagues argue that nearly all taxa observed at the sub-genus level from de-novo OTU clustering or sequence variant detection algorithms are results of sequencing error and that the relationship between taxa abundance and the number of variants observed within that taxa can be easily modeled to produce better descriptions of richness than naïve algorithms that do not take into consideration sequencing error. The authors imply that algorithms like DADA 2 are not working very well in their attempts to correct for sequencing based error rates and that Qiime based OTU clustering at 99% or 97% do not guard against sequencing error causing spurious OTUs. They propose models that explicitly consider rates of sequencing error as correctives. The paper is quite clearly written and both compelling and convincing. It makes an important contribution to our understanding of how classical ecological metrics of species diversity interact with modern sequencing technology.

> We thank Prof. Fodor for their extremely positive comments on this work.

In some ways, the paper refutes early papers in next-generation sequencing metagenomics such as the PNAS paper in 2006 (<https://www.pnas.org/content/103/32/12115/>; "Microbial diversity in the deep sea and the underexplored "rare biosphere"") that argued that there was a "rare biosphere" that represented "a nearly inexhaustible source of genomic innovation". Through a modern understanding such as the one the authors present, it seems we can better understand the observed diversity in such early works as accumulation of sequencing error governed by simple Poisson processes and not related to underlying biology. The data presented in Fig. 2 in the manuscript under review are very compelling in demonstrating beyond doubt that observed richness is a simple and essentially strict function of abundance of the total genus relative abundance. These observations also seem highly consistent with recent observations that the number of variants in 16S datasets can be very well modeled by Poisson sampling taking into account the abundance of each separate variant (as seen in Fig. 2 here: <https://journals.asm.org/doi/full/10.1128/mSystems.00697-21> HashSeq: a Simple, Scalable, and Conservative De Novo Variant Caller for 16S rRNA Gene Data Sets). The authors may wish to consider citing this recent paper (full disclosure, your reviewer here is Anthony Fodor, last author of that paper).

> We thank Prof. Fodor for this discussion. We apologize for our oversight in not citing the closely related HashSeq paper. We have now incorporated this citation in the second paragraph of the Discussions section, where we discuss related literature as below.

Finally, consistent with our results on spurious taxa accumulations, Fouladi et al.,[82] demonstrate the abundance dependent accumulations of 16S sequencing error variants in microbiome datasets.

Overall, the paper is meticulous in its execution and arguments and a model of clarity in presenting complex ideas. If the paper has a limitation, it is that the description of the application of Prokounter to previous datasets could perhaps be a bit clearer and more compelling. In line 238,

for example, the authors write that "Prokounter produced negative richness inferences for both infection and post-infection samples consistent with antibiotic exposure." But does "negative" here mean a negative result or a direction of correlation that is negative?

> We apologize for the confusion in our language. By "negative", we meant "reduced richness". All such confusing instances in Section 2.3 of the manuscript has now been revised.

This section could be made more clear perhaps by showing a figure with a richness difference analyzed "traditionally" and with Prokounter. Likewise, similar figures could be made for the Pathogenesis, Wastewater treatment and Pseudomonas datasets.

> Our strategy with the Applications section was to (1A) demonstrate with figures that *genus-specific* richness estimates are confounded by genus abundances with all existing methods in all datasets, (1B) identify a few examples cases in known / well-controlled settings (the Pseudomonas dilution experiment) where *genus-specific* inferences can be concretely discussed, and finally, (2) assess when *sample-wide* richness inferences deviate (long-term time series study).

Because the Pseudomonas dilution study serves as a concrete practical example for all the above points, as Reviewer #2 recommended, we have moved it to the beginning of the section. To demonstrate point 1A above concretely with all methods and all datasets, we had included two supplementary figures (figure #S8, S12), but had only referenced Fig. S12 it in the beginning of the Section 2.3. We have now explicitly changed this reference as: "(Fig. 2C, S12, compare to Fig. 2B, S8)"

A visual demonstration of the impact of the author's approach on "real world" inference problems in comparison to classical, but naive, approaches might make the paper more accessible for less mathematical readers.

> Thank you, we have additionally incorporated new methods comparisons as suggested by Reviewer #2 (new Figures S21, S22, revised new panel C in Fig. S6).

We thank Prof. Fodor for their service in reviewing this manuscript.

Response to Reviewer #2.

In microbiome studies, the total number of species present (richness) is a key ecological parameter for microbial communities, but is notoriously difficult to measure accurately. The number may be underestimated if not all species are sampled, but more often is overestimated due to errors in PCR amplification and sequencing. The authors of this paper present a new method to estimate richness by breaking the problem down and estimating the richness of defined bacterial groups, rather than the full ensemble. With this basic idea, the authors cleverly recognize that the number of spurious species introduced from sequencing error will increase with the relative abundance of any real species present.

The method is original, clever, important, and will be valuable to researchers in the field.

> We thank the reviewer for their encouraging comments.

However, the manuscript is hampered by two major conceptual issues that prevent it from presenting a clear and solid argument for the approach.

1. "Sub-genus taxa."

[1.1] The authors have placed an enormous amount of emphasis on the taxonomic rank of genus as their primary organizing structure for bacteria. This emphasis is misplaced. The generation of spurious 16S sequences is a purely numeric phenomenon that has to do with PCR errors and fidelity of sequencing instruments. In contrast, a genus is a taxonomic construct invented to name organisms, which is determined using phylogenetic arguments rather than sequence similarity. It just so happens that the typical size of a genus in sequence space coincides approximately with a convenient boundary beyond which spurious species are unlikely to be generated by technical error. The authors might be able to "get away with" using bacterial genera to fuel their software, but I would argue that it is conceptually inappropriate and a gigantic missed opportunity.

[1.2] The idea of a bacterial genus is fundamentally mismatched with the idea of a group of similar 16S sequences that are generated by technical errors. New genera (in modern times) are established using phylogenetic arguments, which are mismatched with the decidedly non-phylogenetic nature of PCR errors. Genera are imperfect--*Escherichia/Shigella* is the classic example, but the examples of *Eubacterium* and *Clostridium* should scare the heck out of the authors. Genera are incomplete--should the lack of well-established genera for mouse intestinal bacteria (S24-7/*Muribaculum*) and segmented filamentous bacteria (SFB) hold this tool back from accurate estimates of richness?

[1.3] The method presented here should be applied to sequence groups that are determined numerically, in a way that matches the process by which sequencing errors are generated. The authors have already presented the solution in their paper, when they explore "taxa accumulations at various taxonomic levels" (line 170). This argument confused me at first, but I think it's a key insight from the study. If the sequence grouping is too broad, you still recover the correlation between relative abundance and species richness. So there is nothing special about the genus

level! It seems that your groupings must be sufficiently broad to capture the error-generated species along with the true species, but narrow enough that the relative abundance can be used to back out the numbers you need.

[1.4] I do want to recognize that using the language of "genus-level" similarity makes for a catchy and easy-to-understand results section. If you are able to define "genus-level" similarity in some numerical way, I think you would be safe to state the definition and then stick with your current language.

> Our methodology derives controls for spurious taxa accumulation based on sub-genus taxa accumulations. Appropriately, the reviewer notes that sub-genus taxa are not necessarily equivalent to erroneous taxa generated by sequencing (paragraph [1.1]), that genus-level annotations may themselves be ill-defined or incomplete (paragraph [1.2]), and identifies that a solution to these concerns already exists in the paper (paragraph [1.3]). Finally, the reviewer suggests that a numerical definition of genus-level similarity will add value in arguing for the generality of the proposed procedure (paragraph [1.2], [1.3] and [1.4]). We address these thoughtful comments below.

While we never claimed exact equivalence between the set of sub-genus taxa in a given genus, and the set of error variants generated by sequencing noise, we now realize that a lack of discussion in contrasting the two, especially along the lines of the reviewer's comments, can confuse the reader, and is a missed opportunity in discussing the generality of the method. Our use of "Genus" category to derive controls for false taxa accumulation was motivated by multiple lines of evidence presented in Results 2.1 that most sub-genus taxa in 16S surveys are likely spurious.

As the reviewer indicates in paragraph [1.3], our results indicate that the false taxa accumulation rate at the Genus level is already quite strong that it overwhelms richness statistics at any arbitrary higher order taxonomic rank. Thus, except in rare situations when genus-specific technical effects that may need to be modeled (Fig. S18), as the reviewer notes, there is nothing special about the Genus level if one is able to estimate the controls necessary when making richness inferences for any arbitrary taxa collection. Indeed, when multiple genera are grouped together (e.g., *Escherichia/Shigella*) due to a lack of 16S gene sequence resolution (paragraph [1.2]), one only needs to shift along the recovered abundance axis to derive the "merged genera" abundance dependent trend based adjustment factor.

In principle, we may lose power though in resolving any genus-specific technical effects (as illustrated in Fig. S18), and this necessitates a discussion of two particular cases. First, if grouping of genera is only made because of very similar/identical 16S sequences, there may not be a need to disentangle genus specific technical effects as spurious error variants are dependent on nucleotide sequence context. Second, when taxonomic descriptions for sequences are incomplete i.e., not available, based on the comprehensive work of Yazra et al., 2014 on rational taxonomic boundaries [REF1], we borrow on the empirically observed numerical 16S sequence

identities for the Genus level, and recommend a numerical measure of 94.5% sequence identity (paragraph [1.4]) for grouping arbitrary collections of 16S sequences.

[REF1] Yarza, P. *et al.* Uniting the classification of cultured and uncultured bacteria and archaea using 16S rRNA gene sequences. *Nature Reviews Microbiology* 12, 635–645 (2014).

We have therefore revised our Discussion section, incorporating elements of the reviewer's comments above and our response, in the following Discussion paragraph (now as paragraph #8):

Emphasis on sub-genus taxa. *We have exploited the characteristics of sub-genus taxa accumulations to estimate controls for the generation of spurious taxa in a microbiome survey. However, our results indicate that estimation of the needed controls could also be done based on higher order taxonomic groupings as they too exhibit similar taxa accumulation characteristics (Fig. 1). In principle, we may lose power in resolving genus-specific technical effects (as illustrated in Fig. S18), and this necessitates a discussion of two cases. First, if grouping of genera is made because of very similar/identical 16S sequences, we may not need to account for individual genus-specific effects within the grouping, as sequencing errors are strongly dependent on nucleotide sequence context. Second, when taxonomic descriptions for sequences are incomplete or are not available, based on the comprehensive work of Yarza et al., 2014 on rational taxonomic boundaries [43], we recommend a numerical measure of 94.5% sequence identity for typing arbitrary collections of 16S sequences to achieve a reasonable "Genus-level" grouping, for estimation purposes.*

2. Denoising

[2.1] Any discussion of denoising methods is conspicuously absent from your paper. You use DADA2 to generate sequence variants, and the denoising process used by DADA2 is supposed to solve exactly the problem you're trying to solve, but you say nothing about it. Ideally, I would run my denoising software, it would characterize all the sequencing errors and remove all the spurious variants, then return a table of authentic 16S sequences. I mean, we all know that's not how it works, but you should explain why you are taking a post-denoising regression/optimization approach, rather than trying to write a denoiser yourself.

[2.2] One might argue (not me, but someone else) that your problem should be solved by the denoiser. Why is it not solved by current denoisers? Is it best to even approach the problem in this way? What is the relationship between your software and denoising software?

> We address these helpful comments in paragraphs [2.1] and [2.2] below.

We had not discussed denoisers in our paper as our focus was on estimating differential richness in the presence of species misclassification (paragraph [2.1]). We would like to note that bulk of supplementary note 1 is devoted to studying the denoising properties of sequence similarity

threshold methods (paragraph [2.1]). We had also indicated the practical difficulties in arriving at an accurate error model of sequencing in the first paragraph of the Results section and the ninth paragraph (previously 8th) of the Discussions sections (paragraphs [2.1-2.2]).

However, as the reviewer notes, we now realize an explicit discussion on the topic can place the method in a solid context, in the backdrop of existing denoisers, a key field of research in 16S bioinformatics. We have therefore added the following points to Discussions paragraph 5:

Unlike human gene expression studies that enjoy a known nucleotide reference catalogue to reconstruct gene expression, microbiome studies often lack a well-defined reference. Thus, microbiome studies typically reconstruct the target microbial population bioinformatically by clustering 16S sequencing reads for sequencing noise, with error models derived based on sequence similarity thresholds, or by modeling sequencing noise. However, arriving at accurate error models is hard in practice (e.g., higher order sequence dependence of errors is not modeled), and thus spurious microbial discoveries are bound to occur when typing several millions of sequencing reads. It is in this context that we find utility with our proposed differential richness inference method, which aims to estimate and correct for the generation of spurious taxa in highly automated micro-ecological surveys.

Minor issues:

1. Straw man richness estimators. As soon as you establish that unique sequences are generated by error, all the assumptions justifying the Chao1 estimator fly out the window. It is obviously not going to work, so why are you using it in your comparisons? You should compare instead to your real competitors, like Chiu & Chao and most definitely Willis. Chao1 is pretty lame here.

> We felt that comparisons to Chao, ACE and Breakaway are useful for two main reasons. First, we wished to understand the sensitivity of classical richness theory, when an ecological survey is contaminated by species misclassifications. This is a major component of our study. We had not found a similar line of inquiry/a characterization of the confounded inferences, in the ecological literature. Second, as noted in the Discussions section (previously paragraph #5, now #6), the microbiome field seems to have accepted classical richness estimators. Our comparisons then, appropriately demonstrate the unforeseen technical effects one subjects their analysis to, when doing so.

We had noted in the Introduction section of our submission, that the Chiu & Chao method was mostly numerically undefined when applied to microbiome data at lower taxonomic levels, a major question addressed in the paper. As an example, when restricted to genus-level richness calculations where the basic Chao1 estimator was well-defined in the long-term time series study, the Chiu-Chao method was numerically undefined for over 95% of genus-level richness calculations. Although we provided the code for demonstrating this, we had failed to be explicit about it in the Results section. We had not explicitly characterized the method in the Willis paper (also referenced in our introduction) in our manuscript for similar reasons: Firstly, the author

appears to dismiss its applicability for richness estimation for 16S surveys in the first page of the 2016 arXiv submission owing to the large standard errors reported. Secondly, like the Chiu & Chao method, it too suffered from being mostly numerically undefined and/or failed numerical convergence.

Because both Chiu & Chao, and the Willis methods attempt to correct spurious singleton frequencies with higher order frequencies (e.g., doubletons, and tripletons), they are undefined whenever certain combinations of higher order frequencies are not positive. As our results indicate, this is not a rare event.

Thus, we could not compare to these existing methods appropriately in our original submission. Following the reviewer's comment, however, we realize that explicit statements about these methods may still be useful to include in the paper. Thus, we have added figures, and additional text to the results section describing the two methods' performance. The takeaways from these additional comparisons are the following:

- (A) The Chiu-Chao method exhibited a much greater degree of numerically undefined values compared to the basal non-sequencing-noise-aware counterpart (i.e., Chao1). Because the Willis method (`Breakaway_nof1`) shares the same estimation backend as its non-sequencing-noise-aware counterpart (i.e., `Breakaway`), the Willis method too exhibited (A) failed numerical convergence in a large fraction of the genus-specific richness estimation tasks as did `Breakaway`, and (B) presented relatively much larger standard errors, as did `Breakaway`.
- (B) Whenever the two methods (Chiu-Chao and Willis) are numerically well-defined for genus-specific richness estimation, as seen in the new supplementary figures noted below, they correlate extremely well with their basal non-sequencing-noise-aware counterparts, indicating that the same confounding effects carry over. Thus, they did not overcome the differential abundance-differential richness confounding problem, the subject of our paper. The two methods attempt to model spurious singleton frequencies. In practice, with sufficient sampling, the entire range of frequencies can be perturbed by spurious discoveries.

We attach three new supplementary figures (1) illustrating the correlations between the two methods and their basic versions (Fig. S21), (2) simulation results showing that the confounding behavior persists with the two estimators as well (Fig. S6C) and (3) a scatter plot of the genus-wise richness fold changes against differential abundance fold changes, as in Fig. 2B, despite the very low number of points estimated (Fig. S22). Finally, (4) we make note on the inferences in the Applications section of the manuscript, Results 2.3.

We conclude that the issues we have noticed about classical methods also apply to the two specialized versions: Chiu-Chao and the Willis method (`Breakaway_nof1`). These results indicate that the impact and significance of the proposed approach is sizeable.

We have included a results paragraph (section 2.3, 4th paragraph) summarizing these additional analyses:

Richness estimates produced by sequencing noise aware estimators Breakaway_nof1 and Chiu-Chao were well correlated with their basic counterparts Breakaway and Chao1 (Fig. S21), implying that the confounded inferences observed with the latter are likely to be inherited. This was confirmed by simulations (Fig. S6C). Visualizations of the behavior of Breakaway_nof1 and Chiu-Chao estimators (such as in Fig. 2B) for genus-level inferences were not possible as both estimators were either mostly numerically undefined at lower levels of the taxonomy and/or suffered failed numerical convergence (Fig. S22). For example, in the time series study discussed below, restricting to those samples where the basic Chao1 estimates were well-defined, the Chiu-Chao estimator was undefined for over 95% of time. Because both Breakaway_nof1 and Chiu-Chao methods attempt to correct spurious singleton frequencies based on ratios of higher order frequencies (e.g., doubletons, tripletons), estimation issues arise whenever these ratios are not well-behaved. The above results indicate that these situations are not rare in 16S microbiome data.

2. When you write ACE/Betta (line 239), I don't know what the Betta means.

> We have now provided a citation for Betta at the very first usage of the method in the Results section of "**Richness estimators and differential analyses**" as: "Differential richness inferences corresponding to the five estimators were obtained with the R package Betta [26]."

3. "Taken together, our results indicate that most sub-genus taxa in 16S surveys are likely spurious." I think you're being too guarded here: you pretty much know that they are spurious, and you have a pretty good idea about what the exact number is.

> Thank you, we have now made this result stronger by removing the word "likely".

4. The Pseudomonas dilution study is a much cleaner and simpler example, suggest covering this before the other examples in 2.3.

> We agree and we have reorganized section 2.3 accordingly also at the request of reviewer 1.

Software issues:

1. Thanks for making this a real, honest-to-goodness R package. Please finish filling out the DESCRIPTION file.

> We have now appropriately updated the Descriptions file with information on URL, BugsReports, Imports and Artistic-2.0 License.

2. Your dependencies should be in the DESCRIPTION, not sprinkled inside your functions. That way, the dependencies get installed automatically when your software is installed. <https://r-pkgs.org/description.html#description-dependencies>

> We have now included dependencies through the “Imports” flag in the Descriptions file.

3. Think carefully about the number of dependencies that go along with your package. MASS and vegan are likely required for basic functionality. Can RColorBrewer and metagenomeSeq be suggested dependencies?

> MetagenomeSeq has now been replaced by the commonly included *Biobase* package.

> We have removed the RColorBrewer dependency.

4. In getProkounterTrends(), you have a keyword argument, plt=FALSE, that returns a plot instead of a data object. This is a common anti-pattern in R libraries. Instead, simply return the data object. Pull that plotting logic out into a second function that takes the data object and returns a plot. Not only will everything be easier to use and test, but it will open some space for you to allow the user to configure the plot.

> We have now separated the plotting routine. The current version always returns the Prokounter Result object, with an option to plot.

We thank Reviewer #2 for their service in their thoughtful and thorough review.

List of changes/clarifications

Manuscript

1. Background

- 1.1. 6th paragraph, which now emphasizes the estimation strategy of the Chiu-Chao method.

2. Results, subsection 2.1

- 2.1. Single colony experiment section, for completeness, along with FST methods' statistics previously included, added the additional statistic "and 90 for a probabilistic clustering method (Dada2)."
- 2.2. Added phrase "suggesting that spurious taxa accumulations overwhelm observed richness statistics" to the end of the penultimate paragraph, to clarify the result about higher taxonomic groupings, as noted by reviewer #2.
- 2.3. Last paragraph, changed phrase from "likely spurious" to "spurious".

3. Results, subsection 2.3

- 3.1. A new 5th paragraph describing the results of Chiu-Chao in generating confounded inferences, and estimation difficulties.
- 3.2. The concrete *Pseudomonas* example begins the applications section, instead of being the last.
- 3.3. Like with all other methods, genus-specific and sample-wide inferences from Chiu-Chao and the Willis methods are noted in the *Pseudomonas* example. This further indicates that confounding issues persist with these methods. Like with all other methods, sample-wide inferences for Chiu-Chao and the Willis methods are now also included.

4. Methods section

- 4.1. Included Chiu-Chao and Breakaway_nof1 under richness estimators and differential analyses.

5. Discussions

- 5.1. Additional material to paragraph 5, emphasizing the utility of Prokounter in the context of denoisers.
- 5.2. New paragraph 8 explaining the emphasis and flexibility of estimating trends based on sub-genus taxa accumulation / higher taxonomic groupings.

6. References

- 6.1. Fouladi et al., paper appears as ref. 82.

7. Supplementary information

- 7.1. Three new figures demonstrating Chiu-Chao and Breakaway_nof1.
 - 7.1.1. Figures S21, S22, revised new subpanel C in Fig. S6.

Software

1. Updated Descriptions file
2. Improved getProkounterTrends function for returning the appropriate Results object.
3. Repository has been updated.

Second round of review

Reviewer 2

The authors have thoughtfully addressed all my points, and taught me a few things in the process. I commend them for a much improved paper. Good work on the code, as well.